# Does higher early neonatal mortality in boys reverse over the neonatal period? A pooled analysis from three trials of Nepal

Seema Subedi [ID],[1] Joanne Katz,[1] Daniel Joseph Erchick [ID],[1]
Andrea Verhulst [ID],[2] Subarna K Khatry,[3] Luke C Mullany,[1] James M Tielsch,[4]
Steven C LeClerq,[1,3] Parul Christian,[1] Keith P West,[1] Michel Guillot[2]

[1]International Health, Johns Hopkins University Bloomberg School of Public Health, Baltimore, Maryland, USA
[2]Population Studies Center, University of Pennsylvania, Philadelphia, Pennsylvania, USA
[3]Nepal Nutrition Intervention Project, Sarlahi, Kathmandu, Nepal
[4]Global Health, George Washington University School of Public Health and Health Services, Washington, District of Columbia, USA

**Correspondence to**
Seema Subedi;
ssubedi2@jhu.edu

## ABSTRACT

**Objectives** Neonatal mortality is generally 20% higher in boys than girls due to biological phenomena. Only a few studies have examined more finely categorised age patterns of neonatal mortality by sex, especially in the first few days of life. The objective of this study is to examine sex differentials in neonatal mortality by detailed ages in a low-income setting.

**Design** This is a secondary observational analysis of data.

**Setting** Rural Sarlahi district, Nepal.

**Participants** Neonates born between 1999 and 2017 in three randomised controlled trials.

**Outcome measures** We calculated study-specific and pooled mortality rates for boys and girls by ages (0–1, 1–3, 3–7, 7–14, 14–21 and 21–28 days) and estimated HR using Cox proportional hazards models for male versus female mortality for treatment and control groups together (n=59 729).

**Results** Neonatal mortality was higher in boys than girls in individual studies: 44.2 vs 39.7 in boys and girls in 1999–2000; 30.0 vs 29.6 in 2002–2006; 33.4 vs 29.4 in 2010–2017; and 33.0 vs 30.2 in the pooled data analysis. Pooled data found that early neonatal mortality (HR=1.17; 95% CI: 1.06 to 1.30) was significantly higher in boys than girls. All individual datasets showed a reversal in mortality by sex after the third week of life. In the fourth week, a reversal was observed, with mortality in girls 2.43 times higher than boys (HR=0.41; 95% CI: 0.31 to 0.79).

**Conclusions** Boys had higher mortality in the first week followed by no sex difference in weeks 2 and 3 and a reversal in risk in week 4, with girls dying at more than twice the rate of boys. This may be a result of gender discrimination and social norms in this setting. Interventions to reduce gender discrimination at the household level may reduce female neonatal mortality.

**Trial registration number** NCT00115271, NCT00109616, NCT01177111.

## STRENGTHS AND LIMITATIONS OF THIS STUDY

⇒ Since the neonates were followed at frequent intervals, we could examine the sex differentials in neonatal mortality at more detailed age (0–1, 1–3, 3–7, 7–14, 14–21 and 21–28 days), which have not been seen in other studies.
⇒ Since we used data from three different trials in the same settings, it was appropriate to analyse by pooling the data.
⇒ We could not examine the determining factors for the main result of the study and our discussions are based on the existing literature.

the predominant causes of death being non-infectious.[2 3 5 6] Several factors associated with higher neonatal mortality in boys include intrauterine growth restriction, respiratory distress syndrome, prematurity and birth asphyxia.[7–10] Studies examining immunological differences in animal models have shown that girls have stronger innate and humoral responses to infection, making them better able to fight infection.[11–13] These studies also show that there is an association between sex hormones and immune function, where testosterone in males suppresses the immune system, while oestradiol and progesterone in females improve both the innate and humoral immune responses.[11–13] Males also have higher birth weights than females, leading to a higher risk of complications during delivery and injuries at birth, although in general, low birth weight is associated with higher mortality.[2 11 14–19] Data from HICs show that the mortality in males is higher than females not only during the neonatal period, but also after the neonatal period, through infancy and beyond.[2 20]

In low/middle-income countries (LMICs) with higher neonatal mortality (more than 30 per 1000 live births (LBs)), sex differences in neonatal mortality have been inconsistent. A multicountry study in sub-Saharan Africa

## INTRODUCTION

Since the 1960s, high-income countries (HICs) have reported higher neonatal mortality rates in boys than girls.[1–6] For overall neonatal mortality, boys are at an approximately 20% greater risk of neonatal mortality than girls. These differences are explained primarily as a biologically driven phenomenon with

reported higher neonatal mortality ratios for boys to girls ranging from 1.1 to 1.6.[21] Similarly, an Indonesian study of Demographic and Health Surveys data reported an adjusted sex difference in neonatal mortality of 1.49 times higher in boys.[22] However, a Pakistani study reported an overall sex difference of 0.82, indicating higher neonatal mortality risk in girls.[23]

Separating neonatal mortality into early and late neonatal mortality, the literature from both HICs and LMICs generally shows that boys have higher rates of mortality than girls in the early neonatal period (first week of life).[4 5 19 23 24] The extent of these differences varies by factors like level of neonatal mortality, causes of neonatal mortality and other region-specific factors. However, sex differences in mortality during the late neonatal period have not been consistent.

Evidence from South Asia has suggested that although boys are at higher risk of death in the early neonatal period, this pattern can reverse in the late neonatal period.[23 24] A study by Rosenstock et al, which used one of the datasets of our analysis (Chlorhexidine Study (CHX)), showed there was a reversal in the mortality pattern by sex in Nepal.[19] In the early neonatal period, boys were at 20% higher mortality risk, assumed to be due to biological factors, whereas in the late neonatal period, girls were at a 43% higher mortality risk. This was associated with ethnicity and the gender structure of siblings in the family rather than by gender preference alone, where girls born to families with only girls had higher risk.[19] In an urban Pakistani study, where overall neonatal mortality was lower in boys (0.82), the sex differences in early and late neonatal mortality were 1.21 and 0.28, respectively, indicating a reversal of risk in the later weeks of the neonatal period.[23] Differential healthcare-seeking behaviours and gender preference for male infants have been reported as explanations for higher late neonatal and infant mortality in girls.[23–28]

Some South Asian studies have examined sex differences in post-neonatal mortality. An analysis of data from a randomised trial in rural northern India comparing sex differences in mortality during the neonatal period and beyond showed that boys had 1.25 times higher neonatal mortality in the first week of life. In the post-neonatal period, however, girls had significantly higher mortality; 1.4 and 1.7 times higher in days 29–180 and days 181–365, respectively.[15] Factors associated with excess female mortality in the post-neonatal period were caste and mother's occupation (higher for mothers working outside the home).

A recent study with 297 509 LBs in India and Pakistan showed that both overall and early neonatal mortality risk were significantly higher in boys than girls. However, there was no significant difference by sex in late neonatal mortality, and mortality between 29 and 42 days.[7]

Given that some South Asian countries showed a reversal in neonatal mortality, and others have not, we examined data from three sequential, community-based randomised controlled trials (RCTs) conducted in the

district of Sarlahi located in the east-central, southern rural plains (Terai) of Nepal. The district is in the rural low-lying area of Nepal that borders the Indian state of Bihar. This area has two main ethnic groups, Pahadi or people of hill origin, and Madeshi, who are from the plains. Health indicators and access to care have changed from 1999 through 2017. In the first trial, only 5% of women delivered in a facility, 9% in the second trial (2002–2006) and 42% in the third trial (2010–2017)[29 30] (Katz, personal communication). This increase in facility delivery coincided with a government cash incentive scheme that paid women to get four antenatal care visits and delivery in a facility. Maternal literacy increased from 20% to 25% to 32% in these trials, and mean birth weight from 2616 g to 2705 g to 2773 g[29 30] (Katz, personal communication). These studies included frequent in-person follow-up of all live born infants with exact date of deaths, allowing us to analyse sex differences in mortality by more finely categorised ages (0–1 day, 1–3 days, 3–7 days, 7–14 days, 14–21 days and 21–28 days). The objective of this study is to examine sex differentials in neonatal mortality by detailed ages using data in a low-income setting. This can help us pinpoint the age at which the pattern of sex difference in mortality changes or reverses, which could help us plan interventions accordingly.

## METHODS

This is a secondary observational analysis of data from three RCTs. Characteristics of the datasets used in this analysis are provided in table 1. The studies provide pregnancy cohorts from 1999 through 2017. Child follow-up duration ranged from 28 days to 5 years. All studies were community-based RCTs conducted in the same rural community of Nepal by the Nepal Nutrition Intervention Project Sarlahi (NNIPS). The first study, NNIPS-3, followed pregnancies and births in the study area from 1999 to 2000, to look at the effect of antenatal multiple micronutrient supplementation on birth outcomes and the health of their children.[29] The second, the CHX Study, followed participants from 2002 to 2006 to assess if a chlorhexidine body wipe and/or chlorhexidine application to the umbilical stump reduced neonatal mortality.[30] The third, the Nepal Oil Massage Study (NOMS), followed participants from 2010 to 2017 to evaluate the impact of sunflower versus mustard seed oil massage on neonatal mortality. In each study, vital status of newborns at birth and through 28 days of life was ascertained in a prospective follow-up done by study teams and date of death was recorded. In the NNIPS-3 and CHX Studies, LBs versus stillbirths were self-reported by mothers. In NOMS, infants were considered born alive if the baby moved, cried or breathed after the birth.

Individual-level data for each pregnancy and LB included date and type (LB/stillbirth) of outcome, date of death, length of follow-up, sex of the infant, and whether the birth was a singleton or multiple. Only LBs were used for this analysis. For each individual dataset, we

**Table 1** Methodologies of studies included in the analysis of sex-specific mortality by age

| Study | Study design | Birth cohorts | Total follow-up time | Neonatal follow-up visits | Neonatal-level intervention | Total LB in analysis | Number of neonatal deaths | Cumulative neonatal mortality (per 1000 LBs) |
|---|---|---|---|---|---|---|---|---|
| NNIPS-3 | Randomised controlled community trial | 1999–2000 | 28 days* | Days 1, 2, 3, 4, 5, 6, 7, 8, 9, 10, 17, 24 and 31 | None | M=2082 F=2045 Total=4127 | M=92 F=81 Total=173 | M=44.2 F=39.7 Overall=41.9 |
| CHX Study | Randomised controlled community trial | 2002–2006 | 28 days | Days 1, 2, 3, 4, 6, 8, 10, 12, 14, 21 and 28 | Chlorhexidine and placebo wipe | M=12 188 F=11 456 Total=23 644 | M=371 F=338 Total=709 | M=30.5 F=29.6 Overall=30.0 |
| NOMS | Randomised controlled community trial | 2010–2017 | 28 days | Days 1, 3, 7, 10, 14, 21 and 28 | Sunflower oil and mustard oil massage | M=16 533 F=15 425 Total=31 958 | M=548 F=449 Total=997 | M=33.4 F=29.4 Overall=31.4 |
| Total | | | | | | 59 729 | 1879 | 31.6 |

*In NNIPS-3, follow-up went beyond 28 days, but for the other two studies, it went through 28 days only.
CHX, Chlorhexidine Study; LB, live birth; NNIPS, Nepal Nutrition Intervention Project Sarlahi; NOMS, Nepal Oil Massage Study.

calculated survival times for LBs using dates of birth and death. Survival times were split into age categories (0–1 (first 24 hours), 1–3, 3–7, 7–14, 14–21 and 21–28 days). The total deaths and person-time in each category were used to calculate death rates, and probability of dying with 95% CI in those groups, separately for boys and girls. Differences in the probability of dying between boys and girls were visualised using mortality curves. Cox regression was used to estimate Hazard Ratios (HRs) with 95% CI for male versus female mortality for overall neonatal mortality (0–28 days), early neonatal mortality (0–7 days), late neonatal mortality (7–28 days) and for the more finely categorised age groups described above. No covariates other than sex were included in the Cox regression. No adjustments were made for time-trends in neonatal mortality rates. The aim of this analysis was not to explain drivers of neonatal mortality trends but rather to compare the differential neonatal survival by sex within the same time periods. Datasets were then pooled to conduct the same analyses. Data were analysed by combining intervention and control groups after fitting a Cox regression model for each study and a pooled model, with an interaction between sex and a binary intervention indicator, which found no significant interaction effects (HR 0.76, 95% CI 0.37 to 1.55; HR 1.08 (0.80 to 1.47); HR 1.14 (0.89 to 1.47); HR 1.07 (0.89 to 1.29)) for NNIPS-3, CHX, NOMS and pooled analysis, respectively.

### Patient and public involvement

Patients or the public were not involved in the design, conduct, reporting or dissemination plans for this research.

### RESULTS

The overall neonatal mortality risk was higher in boys than girls in the individual studies as well as the pooled analysis (N=59 729 LBs) (table 1). Neonatal mortality was 41.9 per 1000 LBs (44.2 vs 39.7 in boys and girls), 30.0 per 1000 LBs (30.5 vs 29.6 in boys and girls), 31.4 per 1000 LBs (33.4 vs 29.4 in boys and girls) and 31.6 per 1000 LBs (33.0 vs 30.2 in boys and girls) in 1999, 2002, 2010 and the pooled analysis, respectively. Child's sex was missing for a very small number of neonatal deaths (1 of 174 in NNIPS-3, none in CHX Study and 4 of 1001 in NOMS).

The 1999 NNIPS-3 Study found that more boys than girls died early (0–1 day, 1–3 days), then the rates for boys and girls converged, until a reversal was seen in the fourth week of life (figure 1) (HR=0.39; 95% CI: 0.08 to 2.03). The 2002 CHX Study found boys had a higher mortality than girls in the early neonatal period (0–1 day, 1–3 days and 3–7 days), then mortality quickly reversed after the first week and continued until the fourth week (figure 1) (HR=0.49; 95% CI: 0.24 to 0.98). The 2010 NOMS had higher mortality in boys than girls in the early neonatal period (0–1 day, 1–3 days and 3–7 days), then mortality converged in the second and third weeks, followed by a reversal after the third week (figure 1) (HR=0.51; 95% CI:

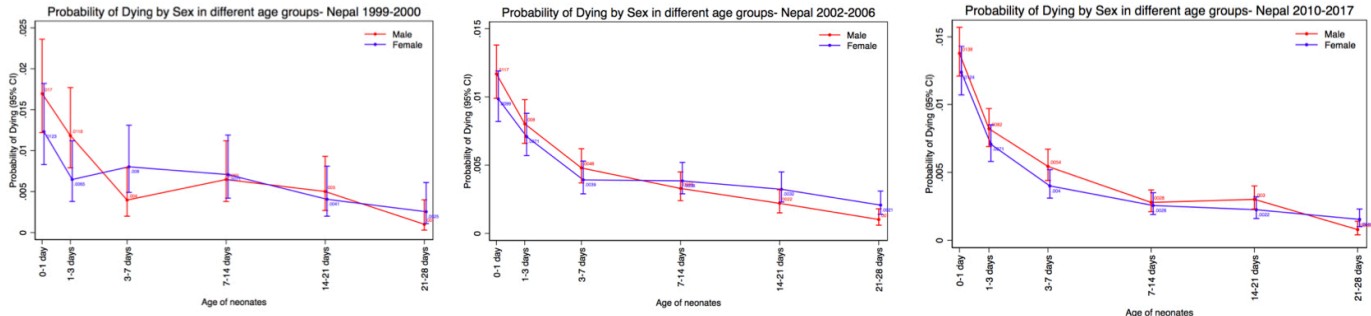

**Figure 1** Sex difference in probability of dying for individual studies: NNIPS-3, CHX and NOMS (from left to right). CHX, Chlorhexidine Study; NNIPS, Nepal Nutrition Intervention Project Sarlahi; NOMS, Nepal Oil Massage Study,

0.25 to 1.04). A common finding in all three studies was that there was a reversal after the third week of life, where female mortality was higher than mortality for boys, although this reversal was statistically significant only in the CHX Study (table 2). Our pooled analysis showed mortality among boys was higher through the second week (0–1 day, 1–3 days, 3–7 days and 7–14 days), followed by similar rates during the third week (14–21 days), followed by a statistically significant reversal in the fourth week of life (21–28 days) (figure 2). For the pooled analysis, results from Cox regression showed that early neonatal mortality (HR=1.17; 95% CI: 1.06 to 1.30) was significantly higher in boys than girls, and the fourth week mortality reversed with 2.43 (95% CI: 1.26 to 3.33) times higher in girls than boys (HR=0.41; 95% CI: 0.31 to 0.79) (table 3). The details of the mortality rates by age and sex, including age group, deaths, person-year, death rate and probability of dying for NNIPS-3 Study, CHX Study, NOMS and pooled analysis, are shown in the online supplemental tables 1–4, respectively.

## DISCUSSION

Our study found higher mortality in boys than girls early in the neonatal period followed by a reversal in the fourth week of life in each of the individual studies. This work extends that of Rosenstock et al to include a longer time span from 1999 through 2017 in the same geographical area.[19] In the pooled analysis, this reversal was statistically significant and the mortality HR was 2.43 times higher in girls than boys in the fourth week of life. This is similar to the findings by Rosenstock et al in rural Nepal and Jehan et al in urban Pakistan although they compared only early and late neonatal mortality and found a reversal in the late neonatal period.[19 23] However, a recent study in India and Pakistan showed higher male mortality in the early neonatal period, but no significant difference in male and female mortality in the late neonatal period or between 29 and 42 days.[7] In that study, the first follow-up was within 48 hours of delivery, with one more visit at 42 days post partum.[7]

**Table 2** HR (male/female) for neonatal mortality for individual studies

| Age category | NNIPS-3 (1999–2000) N=4127 | | | CHX Study (2002–2006) N=23644 | | | NOMS (2010–2017) N=31958 | | |
|---|---|---|---|---|---|---|---|---|---|
| | HR (M/F) | 95% CI | P value | HR (M/F) | 95% CI | P value | HR (M/F) | 95% CI | P value |
| Overall neonatal (0–28 days) | **1.11** | 0.83 to 1.51 | 0.458 | **1.03** | 0.89 to 1.20 | 0.652 | **1.14** | 1.01 to 1.29 | 0.037 |
| Early neonatal (0–7 days) | **1.22** | 0.85 to 1.75 | 0.272 | **1.17** | 0.99 to 1.40 | 0.067 | **1.16** | 1.02 to 1.34 | 0.029 |
| Late neonatal (7–28 days) | **0.91** | 0.53 to 1.57 | 0.744 | **0.71** | 0.53 to 0.96 | 0.024 | **1.04** | 0.79 to 1.38 | 0.777 |
| Subanalysis by days from birth | | | | | | | | | |
| 0–1 day | **1.37** | 0.82 to 2.30 | 0.224 | **1.18** | 0.92 to 1.52 | 0.18 | **1.11** | 0.92 to 1.35 | 0.284 |
| 1–3 days | **1.82** | 0.93 to 3.58 | 0.081 | **1.13** | 0.84 to 1.52 | 0.415 | **1.16** | 0.90 to 1.50 | 0.247 |
| 3–7 days | **0.49** | 0.21 to 1.16 | 0.105 | **1.22** | 0.82 to 1.81 | 0.316 | **1.35** | 0.98 to 1.89 | 0.068 |
| 7–14 days | **0.91** | 0.43 to 0.95 | 0.819 | **0.85** | 0.55 to 1.32 | 0.482 | **1.08** | 0.70 to 1.68 | 0.705 |
| 14–21 days | **1.23** | 0.49 to 3.12 | 0.658 | **0.68** | 0.41 to 1.13 | 0.135 | **1.33** | 0.86 to 2.10 | 0.198 |
| 21–28 days | **0.39** | 0.08 to 2.03 | 0.267 | **0.49** | 0.24 to 0.98 | 0.046 | **0.51** | 0.25 to 1.04 | 0.063 |

CHX, Chlorhexidine Study; NNIPS, Nepal Nutrition Intervention Project Sarlahi; NOMS, Nepal Oil Massage Study.

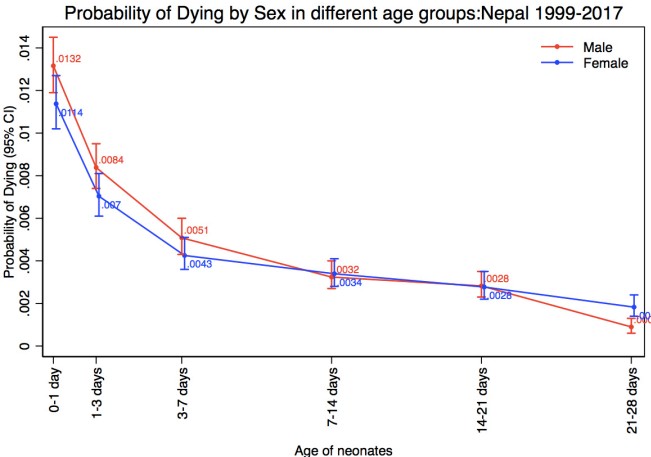

**Figure 2** Sex difference in probability of dying for pooled analysis, Nepal 1999–2017.

Our analysis had more frequent visits and a prospective record of exact date of death.

Another large RCT in rural north India, which examined differences in the post-neonatal period, found a reversal in 29–180 and 181–365 days after birth.[15] They followed LBs on day 29 after the infant's birth and at ages 3, 6, 9 and 12 months to obtain vital status of the infant.[15] For this reason, they analysed the difference in the post-neonatal period but could not do so in the late neonatal period. Our study with rigorous follow-up within the neonatal period allowed us to examine the sex differences in more finely categorised ages and identify that the reversal took place as early as the fourth week of life.

Our previous work with the 2002–2006 dataset further examined possible reasons for sex differences in mortality, finding that ethnicity, differential neonatal care-seeking behaviour and prior family composition with multiple daughters were important factors associated with higher late neonatal mortality in girls.[19 31] While socioeconomic conditions improved over the time period from 1999 to 2017, the only major change in healthcare during the time period of the three studies was an increase in facility delivery, particularly in the trial that spanned 2010–2017, due to the government cash incentive programme. However, there was no evidence that the differential survival of neonates by sex varied over this time period. In Nepal, gender discrimination originates at the household level. Since the 1980s, when the World Fertility Surveys first documented evidence of son preference, Nepal has been categorised as a country with a high level of this preference.[32] This practice is still common, as seen in a 2012 survey on 1000 Nepalese men aged 18–49 years showing that the majority (90%) believed that a man with only daughters is unfortunate and not having a son reflects a lack of moral virtue. Nearly half said that a woman's important roles are limited to taking care of her home and cooking for her family. Married women reported that maintaining an income-generating job is precluded by caregiving for small children (32%), lack of permission from other household/family decision-makers (19%) and the workload at home (18%). Only 26% of married Nepalese women reported making independent decisions regarding their own healthcare.[33] Many women are still restricted to the private sphere with unpaid work. Women work for an estimated average of 268 min a day on household chores, whereas men work for only 56 min.[34] This deprives women of quality education, awareness and exposure. Socially, sons are given preference because of the various cultural and economic roles that are believed to be performed by sons only: performing lighting of the funeral pyre, continuing the family lineage and providing

**Table 3** Probability of dying and HR (male/female) of neonatal mortality for pooled analysis

| Pooled (1999–2017) N=59 729 | | | | | | | | | |
|---|---|---|---|---|---|---|---|---|---|
| | **Boys (N=30 803)** | | | **Girls (N=28 926)** | | | | | |
| Age category | Deaths | Person-year | Probability of dying | Deaths | Person-year | Probability of dying | HR (M/F) | 95% CI | P value |
| Overall neonatal (0–28 days) | 1011 | 826 603 | 0.0086 | 868 | 780 341 | 0.0078 | 1.09 | 1.00 o 1.20 | 0.045 |
| Early neonatal (0–7 days) | 806 | 210 476 | 0.0268 | 646 | 198 355 | 0.0228 | 1.17 | 1.06 to 1.30 | 0.002 |
| Late neonatal (7–28 days) | 205 | 616 127 | 0.0023 | 222 | 581 986 | 0.0027 | 0.87 | 0.72 to 1.05 | 0.158 |
| Subanalysis by days from birth | | | | | | | | | |
| 0–1 day | 401 | 30 472 | 0.0132 | 326 | 28 660 | 0.0114 | 1.15 | 1.00 to 1.34 | 0.051 |
| 1–3 days | 253 | 60 356 | 0.0084 | 200 | 56 846 | 0.0070 | 1.19 | 0.99 to 1.43 | 0.064 |
| 3–7 days | 152 | 119 649 | 0.0051 | 120 | 112 849 | 0.0043 | 1.19 | 0.94 to 1.52 | 0.146 |
| 7–14 days | 96 | 207 511 | 0.0032 | 95 | 195 939 | 0.0034 | 0.94 | 0.72 to 1.27 | 0.745 |
| 14–21 days | 83 | 205 818 | 0.0028 | 77 | 194 441 | 0.0028 | 1.01 | 0.75 to 1.39 | 0.909 |
| 21–28 days | 26 | 202 798 | 0.0009 | 50 | 191 606 | 0.0018 | 0.41 | 0.31 to 0.79 | 0.003 |

old age economic security for their parents, whereas girls are considered an economic liability because they have to live in their parents' home until marriage, for which a dowry must be provided to the groom's family. Upon marriage, they become part of the economy of the husband's family, hence being an economic drain on the family from birth onwards.[35] Nepal's patrilineal and patrilocal social structure combined with socioeconomic and religious values leads to son preference and gender discrimination.

A systematic review found higher care-seeking behaviour for male than female neonates in 17 studies in South Asia, particularly in households with older female siblings.[36] In addition, for male babies, care-seeking was more frequent, from better-qualified care providers, and with higher expenditure compared with girls. Studies also have consistently shown that households with female children were more likely to report discrimination, because family members perceived that care for illness was not so important, leading to reduced care-seeking.[36] Similar to Nepal, a 2011 UNICEF report on China also indicated that the discrimination against female infants was highest for those who had older female siblings.[37]

In Nepal, where most children are exclusively breast fed, the median duration of breast feeding for boys and girls was 4.2 months and 4.1 months, respectively.[38] Hence, this is an unlikely explanation for the reversal of mortality in the fourth week of life. The differential may be explained by poorer nutrition, care and rest provided to mothers giving birth to daughters, son preference being the root cause of discrimination in the family. It could also point to specific parental behaviours that take place starting at that age and suggest a critical age window for intervention. However, the 3-week threshold could also just be an indication that the gender discrimination starts from birth or even earlier, but takes at least 3 weeks for the biological and natural survival advantage of females to be overcome by the social advantage of males. Further studies could explore more about why this reversal starts specifically in the fourth week of life.

Neonatal mortality in Nepal has continued to decrease from 39 to 21 per 1000 LBs, with male neonatal mortality decreasing from 52 to 24 per 1000 LBs (reduction of 28%) and female neonatal mortality decreasing from 43 to 17 per 1000 LBs (reduction of 26%) from 2001 to 2016.[33 38] Given that boys have a biological disadvantage in neonatal survival, one could have expected female neonatal mortality to have decreased more than for boys. If female mortality could decrease more than it has, this could contribute to a greater reduction in overall neonatal mortality. Although neonatal mortality in Nepal was decreasing from 2001 to 2016, it still contributed to a higher percentage of under-5 child mortality because mortality among older children has decreased faster than neonatal mortality.[39] If this trend continues, the Sustainable Development Goals (SDGs) target of reducing neonatal mortality to 12 per 1000 LBs by 2030 in Nepal will be difficult to achieve.[40] Therefore, a focus on reducing

female neonatal mortality could help meet the SDG for neonatal mortality and for gender equity. A cross-national study from 138 countries also showed evidence that the Gender Inequality Index was positively associated with neonatal mortality.[41] Applying interventions to address gender discrimination by addressing cultural and social barriers at the household level may help reduce neonatal mortality.

In the early neonatal period, preterm birth is one of the main causes of deaths, while in the late neonatal period, sepsis and pneumonia account for more deaths.[42 43] Studies also show that preterm birth is higher in boys than girls.[44–46] Given the biological susceptibility of boys towards more early neonatal deaths both in the LMICs and HICs, it is not as easy to intervene. However, the main causes of late neonatal mortality like sepsis and pneumonia can be intervened on through improved care-seeking practices. So, if male and female children are provided with similar care-seeking practices in the late neonatal period, the sex differences in neonatal mortality might be reduced, as in HICs.

Gender discrimination not only affects the quality of life of girls and women, but also reduces their survival in the neonatal period. Interventions to strengthen gender equality, such as counselling to woman and their family during antenatal care and postnatal care visits may be helpful to improve female and male neonatal survival. Since the reversal takes place during fourth week of life, specific counselling interventions to parents and family could be targeted in the first 3 weeks of a child's life.

The strength of this study is that it incorporates data from sequential randomised community trials from the same site sharing many similar field procedures carried out by the same highly trained field teams over a 15-year period, so that pooling is reasonable and allows for a more precise analysis of sex differences in neonatal mortality. In addition, these studies have enrolment from pregnancy, which reduces the likelihood that early child deaths have been missed. Neonatal deaths have been tracked at frequent intervals to improve accuracy of age at death, enabling us to conduct survival analysis and Cox regression to obtain improved mortality estimates and HRs.

This study was not able to examine the specific reasons for the mortality reversal. However, we have discussed possible reasons based on the existing literature. Further studies could examine why this reversal takes place as early as the fourth week of life and whether this reversal persists beyond the neonatal period.

## CONCLUSION

Male mortality is higher than female mortality in the early neonatal period, a biological phenomenon seen worldwide. However, this natural pattern is quickly reversed after the third week of life in Nepal. This is likely due to gender discrimination and social norms that operate at household level. Implementing interventions to

reduce gender discrimination at the household level could prevent this reversal and decrease female neonatal mortality, thereby reducing overall neonatal mortality and improving gender equity.

**Acknowledgements** We thank the women, infants and their families who participated in this study, and the NNIPS project staff in Nepal. We are also grateful to Mrs Lee S-F Wu from Johns Hopkins Bloomberg School of Public Health for her contribution to the NNIPS-3 Study.

**Contributors** SS, JK, DJE, AV and MG conceptualised and designed the study. SS conducted the analysis and wrote the manuscript. SKK, LCM, JMT, SCL, PC and KPW were the investigators of the three different trials used in the paper, and along with other authors, they have reviewed results, discussed interpretations, and contributed to development and revision of the manuscript. SS accepts full responsibility for the finished work and the conduct of the study, had access to the data, and controlled the decision to publish.

**Funding** This work was supported by the National Institute for Child Health and Human Development (NICHD 1R01HD090082-01 and R01HD092411). NNIPS-3 was supported by Cooperative agreement (HRN-A-00-97-00015-00) between the Office of Health and Nutrition, US Agency for International Development (USAID), Washington, DC, USA with additional support from the UNICEF Country Office, Kathmandu, Nepal (grant number: NA), and grant GH614 from the Bill and Melinda Gates Foundation (Seattle, Washington, USA). The CHX Study was supported by R01HD044004 and R01HD038753 from the National Institutes of Health (Bethesda, Maryland, USA); grant 810–2054 from the Bill and Melinda Gates Foundation (Seattle, Washington, USA), and Cooperative Agreements (HRN-A-00-97-00015-00 and GHS-A-00-03-000019-00) between Johns Hopkins University and the Office of Health and Nutrition, US Agency for International Development (Washington, DC, USA). NOMS was supported by the National Institutes for Child Health and Development (HD060712) and the Bill and Melinda Gates Foundation (OPP1084399).

**Disclaimer** The funders did not have a role in the design of the study, the data collection, nor the analysis, interpretation and writing of the manuscript.

**Competing interests** None declared.

**Patient and public involvement** Patients and/or the public were not involved in the design, or conduct, or reporting, or dissemination plans of this research.

**Patient consent for publication** Not required.

**Ethics approval** The research proposal (IRB protocol number 827014, and IRB number 8) was considered exempt by the Institutional Review Board (IRB) at the University of Pennsylvania, authorised by 45 CFR 46.101, category 4. NNIPS-3, CHX and NOMS Studies were approved by the IRB of the Johns Hopkins Bloomberg School of Public Health. NNIPS-3 and CHX were approved by the IRB of the Institute of Medicine, Tribhuvan University in Nepal. NOMS was approved by the Nepal Health Research Council in Nepal. Verbal consent was obtained from women for their participation and their infants, for all studies.

**Provenance and peer review** Not commissioned; externally peer reviewed.

**Data availability statement** No data are available. NA.

**ORCID iDs**
Seema Subedi http://orcid.org/0000-0002-6360-3998
Daniel Joseph Erchick http://orcid.org/0000-0002-2852-280X
Andrea Verhulst http://orcid.org/0000-0003-0571-1330

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
