## [Reviewer comments · BMJ Open]

ARTICLE DETAILS

TITLE (PROVISIONAL)	Does higher early neonatal mortality in boys reverse over the neonatal period? A pooled analysis from three trials of Nepal
AUTHORS	Subedi, Seema; Katz, Joanne; Erchick, Daniel; Verhulst, Andrea; Khatry, Subarna; Mullany, Luke C.; Tielsch, James; LeClerq, Steven; Christian, P; West, KP; Guillot, Michel

VERSION 1 – REVIEW

REVIEWER	Michal Simchen Sheba Medical Center, Obstetrics and Gynecology
REVIEW RETURNED	21-Nov-2021

GENERAL COMMENTS	Fascinating manuscript on a very interesting and important topic. The authors describe changes in neonatal mortality pattern by gender in a pooled analysis of data from 3 RCT's conducted on the same rural community in Nepal over a 15 year period. The main findings presented are that although male neonates were at an increased risk of dying over the first 7 days of life compared with females, this trend changes over time and on the 4th week of life female neonates are at a significantly increased risk of dying compared with males. The authors speculate this is due to gender discrimination and different health care seeking behaviour for male and female neonates. I commend the authors on their effort. The manuscript is clearly written and thought provoking. I have several very minor comments: Abstract - p.4 line 35 'pooled data' should be changed to 'pooled data analysis' Methods - p.8 line 30 'breath' should be changed to 'breathed' Table 2 + Table 3 - 'more finely categorized ages' should be changed to 'subanalysis by days from birth'
---

REVIEWER	Lena Karlsson Umeå Universitet, Demographic and Ageing Research
REVIEW RETURNED	06-Dec-2021

GENERAL COMMENTS	2. Minor: There is an inconsistency of the years for the individual datasets expressed in the abstract vs tables/figures and in the text. In the abstract the intermediate period is 2002-2005, in tables/figures and the following text, the period is 2002-2006. 3. The manuscript lacks a clear statement of the main study aim, and research questions. At page 6, the authors to a higher extent explain the possibilities of the datasets – “allowing us to analyze sex differences in mortality by more finely categorized ages”. I think that that the study aim and research questions should be stated first, followed by a description of the possibilities to conduct the study (in
---

	accordance to the aim and questions). The authors explain the study design – “three community-based randomized controlled trials conducted in rural Nepal” and gives a brief description of the study region: “the District of Sarlahi located in the east-central, southern rural plains (Terai)”. I would suggest that the authors add some more information about maternal and neonatal health care in the study area/region covering the period 1999-2017, for example, percentages of births in hospitals vs home assisted births, development of maternal, obstetric, and neonatal health care over the period. Further, I suggest adding a short description about the ethnical composition of the population and vital socioeconomical and political changes during the study period (1999-2017) that might influence the results (neonatal mortality rates, sex differences etc). The latter could be included in the Discussion section. 7 & 12. My major concerns regarding the way the pooled data is used in this study. From my point of view there are several shortcomings using pooled cross sections from different time-periods and that they should be handled/considered or discussed to a higher extent. First, there is the question whether the same model apply in each time-period? Second, if there should be adjustments for time-trends in neonatal mortality rates? Third, where there any important changes in contextual factors between the time-periods, for example health care systems, socioeconomical and political factors, that might have influenced the sex differences in late neonatal mortality (21-28 days)? Following the results presented for the three individual datasets (Table 2); the sex difference in late neonatal mortality (21-28 days) is only significant for the intermediate period (2002-2006). By pooling the data – and thus having more “power”- hides the differences between each time-period. My suggestion is to include Table 2 in an appendix, include time dummies in Table 3, and to a higher extent discuss the results in relation to the population sizes and trends in sex differences in late neonatal over time etc. As I perceive, that should give a more correct picture of the sex differences in late neonatal mortality (21-28 days) over the entire study period. 10. The titles of Table 2 should include that the HRs are for boys. Figure 2 should be enlarged.
--	---

VERSION 1 – AUTHOR RESPONSE

Reviewer: 1

Prof. Michal Simchen, Sheba Medical Center Comments to the Author:

Fascinating manuscript on a very interesting and important topic. The authors describe changes in neonatal mortality pattern by gender in a pooled analysis of data from 3 RCT's conducted on the same rural community in Nepal over a 15 year period. The main findings presented are that although male neonates were at an increased risk of dying over the first 7 days of life compared with females, this trend changes over time and on the 4th week of life female neonates are at a significantly increased risk of dying compared with males. The authors speculate this is due to gender discrimination and different health care seeking behaviour for male and female neonates. I commend the authors on their effort. The manuscript is clearly written and thought provoking. I have several very minor comments:

1. Abstract - p.4 line 35 'pooled data' should be changed to 'pooled data analysis'

Thank you very much for your encouraging words above. Since the abstract was changed substantially, we now say “pooled data analysis” within the first sentence of the Results section of the abstract.

2. Methods - p.8 line 30 'breath' should be changed to 'breathed'

Done

3. Table 2 + Table 3 - 'more finely categorized ages' should be changed to 'subanalysis by days from birth'

This has been changed to “sub-analysis by days from birth” in Tables 2 and 3.

Reviewer: 2

Dr. Lena Karlsson, Umeå Universitet

Comments to the Author:

1. Minor: There is an inconsistency of the years for the individual datasets expressed in the abstract vs tables/figures and in the text. In the abstract the intermediate period is 2002-2005, in tables/figures and the following text, the period is 2002-2006.

Thank you for identifying this typo. The date is changed to 2002-2006 in the abstract.

2. The manuscript lacks a clear statement of the main study aim, and research questions. At page 6, the authors to a higher extent explain the possibilities of the datasets – “allowing us to analyze sex differences in mortality by more finely categorized ages”. I think that that the study aim and research questions should be stated first, followed by a description of the possibilities to conduct the study (in accordance to the aim and questions).

Thank you. We have now added “The objective of this study is to examine sex differentials in neonatal mortality by detailed ages using data in a low-income setting” as the last sentence of the “Objectives” section of the abstract and also at the end of the Introduction in the main text we have now added “The objective of this study is to examine sex differentials in neonatal mortality by detailed ages using data in a low-income setting. Examining the sex differential in mortality by more detailed ages can help us pinpoint the age at which the pattern of sex difference in mortality changes or reverses, which could help us plan interventions accordingly.”

3. The authors explain the study design – “three community-based randomized controlled trials conducted in rural Nepal” and gives a brief description of the study region: “the District of Sarlahi located in the east-central, southern rural plains (Terai)”. I would suggest that the authors add some more information about maternal and neonatal health care in the study area/region covering the period 1999-2017, for example, percentages of births in hospitals vs home assisted births, development of maternal, obstetric, and neonatal health care over the period. Further, I suggest adding a short description about the ethnical composition of the population and vital socioeconomical and political changes during the study period (1999-2017) that might influence the results (neonatal mortality rates, sex differences etc). The latter could be included in the Discussion section.

We have added a description of health indicators and measures of health care access and utilization into the Introduction for each of the three studies. Specifically, we have added to the Introduction: “The district is in the rural low-lying area of Nepal that borders the Indian state of Bihar. This area has two main ethnic groups, Pahadi or people of hill origin, and Madeshi, who

are from the plains. Health indicators and access to care has changed from 1999 through 2017. In the first trial, only 5% of women delivered in a facility, 9% in the second trial (2002-2006), and 42% in the third trial (2010-2017)²⁹⁻³⁰, personal communication (Katz). This increase in facility delivery coincided with a government cash incentive scheme that paid women to get 4 antenatal care visits and delivery in a facility. Maternal literacy increased from 20% to 25% to 32% in these trials, and mean birthweight from 2616g to 2705g to 2773g.”

In the Discussion we have added “While socioeconomic conditions improved over the time period from 1999 to 2017, the only major changes in health care during the time period of the three studies was an increase in facility delivery, particularly in the trial that spanned 2010-2017, due to the government cash incentive program. However, there was no evidence that the differential survival of neonates by sex varied over this time period.”

- 4. My major concerns regarding the way the pooled data is used in this study. From my point of view there are several shortcomings using pooled cross sections from different time-periods and that they should be handled/considered or discussed to a higher extent. First, there is the question whether the same model apply in each time-period?**

The same model was applied to each time period. There were no additional covariates besides sex of the infant. We added to the Methods “No covariates other than sex were included in the Cox regression.” to clarify this.

- 5. Second, if there should be adjustments for time-trends in neonatal mortality rates?**

We have added to the Methods that “No adjustments were made for time-trends in neonatal mortality rates. The aim of this analysis was not to explain neonatal mortality trends but rather to compare within the same time period the differential neonatal survival by sex.”

- 6. Third, where there any important changes in contextual factors between the time-periods, for example health care systems, socioeconomic and political factors, that might have influenced the sex differences in late neonatal mortality (21-28 days)?**

Although many contextual factors changed over this time period, there was no evidence that the sex difference in late neonatal mortality varied over this time period (confidence intervals overlapped for the three hazard ratio estimates). Hence, we do not believe these temporal changes influenced the late neonatal mortality sex differential.

- 7. Following the results presented for the three individual datasets (Table 2); the sex difference in late neonatal mortality (21-28 days) is only significant for the intermediate period (2002-2006). By pooling the data – and thus having more “power”- hides the differences between each time-period. My suggestion is to include Table 2 in an appendix, include time dummies in Table 3, and to a higher extent discuss the results in relation to the population sizes and trends in sex differences in late neonatal over time etc. As I perceive, that should give a more correct picture of the sex differences in late neonatal mortality (21-28 days) over the entire study period.**

We understand the concerns of the reviewer regarding pooling, but believe Table 2 is important to keep in the main paper exactly for the reason the reviewer notes, that individually, only time 2002-2006 has a statistically significant sex difference for 21-28 days. We do not understand what is meant by including time dummies in Table 3. The time differences can be seen in Table 2 and the estimates for the hazard ratios are not different by study/time period in the 21-28 day period. Therefore we believe it is reasonable to pool these data to produce one pooled estimate.

- 8. The titles of Table 2 should include that the HRs are for boys.**

We added (Male/Female) in the title, and similar done for Table 3 as well. “Table 2: Hazard Ratio (Male/Female) for Neonatal Mortality for Individual Studies”.

9. Figure 2 should be enlarged.

Enlarged Figure 2 has been included now.

VERSION 2 – REVIEW

REVIEWER	Lena Karlsson Umeå Universitet, Demographic and Ageing Research
REVIEW RETURNED	21-Mar-2022
GENERAL COMMENTS	I think that the authors have made a great effort in revising the manuscript and replied to all reviewers comments. Minor revision (number 10): I could not find Figure 1 and Figure 2 in the revised manuscript.